# Bacterial Lipid II Analogs: Novel In Vitro Substrates for Mammalian Oligosaccharyl Diphosphodolichol Diphosphatase (DLODP) Activities

**DOI:** 10.3390/molecules24112135

**Published:** 2019-06-06

**Authors:** Ahmad Massarweh, Michael Bosco, Isabelle Chantret, Thibaut Léger, Layla Jamal, David I. Roper, Christopher G. Dowson, Patricia Busca, Ahmed Bouhss, Christine Gravier-Pelletier, Stuart E. H. Moore

**Affiliations:** 1Université de Paris, INSERM U1149, 16 rue Henri Huchard, 75018 Paris, France; a.massarweh@najah.edu (A.M.); isabelle.chantret@inserm.fr (I.C.); leilajamal312@gmail.com (L.J.); 2Université de Paris, CICB-Paris, CNRS UMR8601, Laboratoire de Chimie et Biochimie Pharmacologiques et Toxicologiques, 45 rue des Saints-Pères, 75006 Paris, France; mik_bosco@yahoo.fr (M.B.); Patricia.Busca@parisdescartes.fr (P.B.); Christine.Gravier-Pelletier@parisdescartes.fr (C.G.-P.); 3Mass Spectrometry Laboratory, Institut Jacques Monod, UMR 7592, Université de Paris, CNRS, F-75205 Paris, France; thibaut.leger@ijm.fr; 4School of Life Sciences, University of Warwick, Coventry CV4 7AL, UK; David.Roper@warwick.ac.uk (D.I.R.); C.G.Dowson@warwick.ac.uk (C.G.D.); 5Institute for Integrative Biology of the Cell (I2BC), CEA, CNRS, Univ Paris-Sud, Université Paris-Saclay, 91198 Gif-sur-Yvette, France; ahmed.bouhss@univ-evry.fr; 6Laboratoire Structure-Activité des Biomolécules Normales et Pathologiques (SABNP), Univ Evry, INSERM U1204, Université Paris-Saclay, 91025 Evry, France

**Keywords:** protein *N*-glycosylation, lipid-linked oligosaccharide, congenital disorders of glycosylation, endoplasmic reticulum, peptidoglycan biosynthesis

## Abstract

Mammalian protein *N*-glycosylation requires the transfer of an oligosaccharide containing 2 residues of N-acetylglucosamine, 9 residues of mannose and 3 residues of glucose (Glc_3_Man_9_ GlcNAc_2_) from Glc_3_Man_9_GlcNAc_2_-diphospho (PP)-dolichol (DLO) onto proteins in the endoplasmic reticulum (ER). Under some pathophysiological conditions, DLO biosynthesis is perturbed, and truncated DLO is hydrolyzed to yield oligosaccharyl phosphates (OSP) via unidentified mechanisms. DLO diphosphatase activity (DLODP) was described in vitro, but its characterization is hampered by a lack of convenient non-radioactive substrates. Our objective was to develop a fluorescence-based assay for DLO hydrolysis. Using a vancomycin-based solid-phase extraction procedure coupled with thin layer chromatography (TLC) and mass spectrometry, we demonstrate that mouse liver membrane extracts hydrolyze fluorescent bacterial lipid II (LII: GlcNAc-MurNAc(dansyl-pentapeptide)-PP-undecaprenol) to yield GlcNAc-MurNAc(dansyl-pentapeptide)-P (GM5P). GM5P production by solubilized liver microsomal proteins shows similar biochemical characteristics to those reported for human hepatocellular carcinoma HepG2 cell DLODP activity. To conclude, we show, for the first time, hydrolysis of lipid II by a eukaryotic enzyme. As LII and DLO are hydrolyzed by the same, or closely related, enzymes, fluorescent lipid II analogs are convenient non-radioactive substrates for investigating DLODP and DLODP-like activities.

## 1. Introduction

In mammals, protein *N*-glycosylation is essential for life [1,2], and, in the human population, mutations in genes required for this process underlie a group of rare inherited metabolic diseases called congenital disorders of glycosylation (CDG) [3,4]. As shown in Figure 1, *N*-glycosylation is initiated in the endoplasmic reticulum (ER) by the sequential addition of monosaccharides to dolichyl-phosphate (DP) to yield mature Glc_3_Man_9_GlcNAc_2_-PP-dolichol (DLO) [5,6].

The oligosaccharide moiety is transferred from the lipid carrier onto protein in the lumen of the ER by oligosaccharyltransferase (OST). In certain types of CDG, mutations in the genes required for DLO biosynthesis and transfer of the oligosaccharide onto proteins lead to insufficient DLO production and glycoprotein hypoglycosylation [11]. The mechanisms that regulate DLO biosynthesis in either normal cells or cells from these CDG patients are poorly understood, but DLO hydrolysis, yielding either dolichyl-diphosphate (DPP) or DP, and either free oligosaccharides (fOS) or oligosaccharyl phosphates (OSP), respectively, could potentially play a role in the control of DLO quality and/or quantity. Mature DLO is hydrolyzed by OST to give rise to fOS in the lumen of the ER, whereas truncated DLO intermediates (Man_7-0_GlcNAc_2_-PP-dolichol), which are seen in cells from patients with certain subtypes of CDG [8,9], and during glucose starvation of normal cells [10], are hydrolyzed to yield cytoplasmic OSP (Figure 1). The molecular machineries involved in OSP generation are yet to be identified at the molecular level. Membranes prepared from *Saccharomyces cerevisiae* possess a Ca^2+^/Mn^2+^-activated OSP generating activity [12], whereas a Co^2+^-activated oligosaccharyl diphosphodolichol diphosphatase (EC 3.6.1.44, DLO diphosphatase (DLODP)) was found in mammalian tissues and cells [13]. The mouse liver DLODP involves microsomal activity that, after centrifugation in density gradients, distributes similarly to an enzyme of the Golgi apparatus (GA) and differently to key, ER-situated enzymes involved in DLO biosynthesis [13]. Whether or not this activity is responsible for OSP generation seen in cells from CDG patients remains to be determined. Definitive identification of the physiological role of this activity is hampered because tools to study OSP generation both in vivo and in vitro do not exist. At present, the only assays for DLODP activities rely on radioactive DLO substrates [12,13] or HPLC/fluorimetry of dephosphorylated OSP after pre-column derivatization [10]. Due to environmental, expense, and time issues, these assays are not appropriate for high-throughput applications such as setting up genetic screens or monitoring column effluent. Our recent data demonstrate that lipid II, the precursor required for bacterial peptidoglycan biosynthesis [14], inhibits the Co^2+^-dependent generation of [^3^H]OSP from [^3^H]DLO [15] by human hepatocellular carcinoma HepG2 cell extracts. These data suggested that bacterial lipid II might be a potential in vitro substrate for DLODP and that dansylated lipid II analogs have the potential to be used as fluorescent substrates for the assay of mammalian and yeast DLODP activities.

In the present study, a simple procedure for measuring the cleavage of fluorescent bacterial lipid II is described, and hydrolysis of GlcNAcβ1,4MurNAc (dansyl pentapeptide)-PP-undecaprenol, to yield GlcNAcβ1,4MurNAc(dansyl pentapeptide)-P, by mammalian microsomal proteins, is demonstrated. The biochemical characteristics of these reactions are similar to those displayed by the reactions required for [^3^H]OSP generation from [^3^H]DLO.

## 2. Results

### 2.1. Generation of Vancomycin Affinity Beads for Solid-Phase Extraction of d-Ala–d-Ala-Containing Compounds

Our previous results demonstrated that bacterial lipid II inhibits the cleavage of [^3^H]DLO by microsomal DLODP [15], suggesting that bacterial lipids and their analogs may themselves be DLODP substrates. In order to examine the potential cleavage of bacterial lipid II by DLODP, we used fluorescent dansylated bacterial lipid II analogs (see Section 4). Direct analysis of reaction mixtures by thin-layer chromatography (TLC) for potential DLODP reaction products was difficult due to interference by buffer components. Therefore, we chose to isolate substrates and potential products from components of the incubation mixtures using vancomycin affinity beads as a solid-phase extraction (SPE) matrix (Figure 2).

Vancomycin is a glycopeptide antibiotic with high affinity for the d-Ala–d-Ala motif of peptidoglycan, peptidoglycan fragments, and bacterial lipids I and II [16,17,18]. Vancomycin, immobilized on 5-Carboxypentyl-Sepharose 4B N-hydroxysuccinimide ester (CH-Sepharose), was used to purify water-soluble peptidoglycan fragments [19], but there are no reports of the use of this technique to extract bacterial LI or LII. CH-Sepharose is no longer commercially available; thus, vancomycin was immobilized using N-hydroxysuccinimide (NHS)-Sepharose as described in Section 4. Next, lipid II was taken up in different concentrations of detergent and incubated with Sepharose-linked vancomycin (Vm-beads) for 1 h at room temperature. The Vm-beads were then washed with the solutions shown in the upper panel of Figure 3A.

When the binding and buffer wash steps were carried out in either 0.2% or 0.02% nonyl phenoxypolyethoxylethanol (NP-40), a fluorescent compound was eluted from the beads with MeOH containing 15 mM NH_4_OH, which, after TLC (not shown), was shown to correspond to lipid II. When 2.0% NP-40 was used during the binding and buffer wash steps, a reduced recovery of lipid II in the MeOH/NH_4_OH eluate, concomitant with higher fluorescence associated with the column run-through fractions, was observed. The binding of water-soluble uridinyl diphospho (UDP)-MurNAc pentapeptide (UDPM5; see Section 4, Table 1) to Vm-beads was much less sensitive to detergent, and this water-soluble compound was eluted using 15 mM NH_4_OH (Figure 3A, lower panel). Lipid II does not bind to Tris-blocked NHS-Sepharose beads (S.M. and L.J., unpublished observations). Data shown in the upper panel of Figure 3B demonstrate that binding of lipid II to Vm-beads is blocked by acetyl-Lys–d-Ala–d-Ala, confirming that the lipid binds to Vm-beads via its d-Ala–d-Ala moiety. Therefore, in the presence of 0.02% NP-40, both UDPM5 and lipid II bind to Vm-beads and, subsequent to washing the beads with H_2_O, the water-soluble and lipophilic compounds can be recovered in the 15 mM NH_4_OH and 15 mM NH_4_OH/MeOH eluates, respectively. Finally, colicin M, which cleaves bacterial lipids I and II to yield undecaprenol and the water-soluble fragments MurNAc(pentapeptide)-PP (M5PP) and GlcNAcβ1,4MurNAc(pentapeptide)-PP (GM5PP), respectively [20], was used to test the utility of the SPE procedure. As expected, incubation of lipid II (Figure 3B, lower panel) with colicin M promoted a reduction of fluorescence associated with the 15 mM NH_4_OH/MeOH eluate and an increase in fluorescence associated with the 15 mM NH_4_OH eluate.

### 2.2. Bacterial Lipid II is Hydrolyzed by Mouse Liver Microsomal Proteins to Yield GlcNAcβ1,4MurNAc(pentapeptide)-P

Next, lipid II was incubated with solubilized mouse liver microsomal proteins, in either the absence or presence of ethylenediaminetetraacetic acid (EDTA), for 0, 2, and 16 h at 37 °C. After SPE, fluorescence associated with the 15 mM NH_4_OH and 15 mM NH_4_OH/MeOH eluates was measured, as shown in Figure 4A.

A time-dependent, EDTA-sensitive decrease in fluorescence associated with 15 mM NH_4_OH/MeOH eluate was accompanied by a time-dependent, EDTA-sensitive increase in fluorescence associated with the 15 mM NH_4_OH eluate. TLC analysis of the fluorescent compounds recovered after SPE of similar incubations, in which LII(DAP)55 was incubated in the absence of EDTA, is shown in Figure 4B. A decrease in LII in the 15 mM NH_4_OH/MeOH eluates was accompanied by an increase in the compound, **a**, recovered in the 15 mM NH_4_OH/H_2_O eluates. A compound with the same mobility as **a** was also detected when LII(Lys)55 was used as substrate (S.M. and L.J., unpublished observations). Whatever the substrate used, small amounts of a slower migrating compound, **b**, were observed. This compound, which was detected maximally in the zero time points, appeared to be quite stable during prolonged incubations (Figure 4B). Next, in order to identify **a**, material recovered in the 15 mM NH_4_OH/H_2_O eluate from an SPE-extracted 16-h incubation of LII with rat liver microsomal protein was analyzed by mass spectrometry in negative ion mode as shown in Figure 5.

The major signal (*z* = 1, 1322.4431) is compatible (mass error: 2.90 ppm) with the presence of GlcNAcβ1,4MurNAc(pentapeptide)-P (GM5P, Figure 5 inset). The observed isotopolog distribution of this component (upper panel of Appendix A) is similar to the theoretical isotopolog distribution (lower panel of Appendix A) that was calculated for GM5P [21]. Signals compatible with the presence of either GlcNAcβ1,4MurNAc(pentapeptide)-PP or GlcNAcβ1,4MurNAc-pentapeptide were not detected (Figure 5). In addition, signals corresponding to compounds missing the residue of *N*-acetylglucosamine (M5PP, M5P, and M5) were not detected (Figure 5). Finally, in order to obtain more structural information about the compound that gave rise to the major signal in Figure 4, MS/MS was performed as shown in Figure 6.

The detection of masses corresponding to losses of phosphate, HexNAc, alanine, and glutamic acid moieties is consistent with the parent compound being GM5P (Figure 5). To conclude, the major product recovered from incubations of LII with rat liver microsomal protein was GM5P.

### 2.3. Bacterial Lipid I and Short-Chain Lipid II Analogs Are Also Hydrolyzed by Liver Microsomal Proteins

Bacterial lipid I (LI) is the biosynthetic precursor of LII and does not possess the terminal non-reducing *N*-acetylglucosamine (GlcNAc) residue of the latter compound. In order to determine whether or not this residue is required for LII hydrolysis by solubilized rat liver microsomal protein, LI digestion products were analyzed by TLC. As shown in Figure 7A (left fluorogram), the migration positions of MurNAc(pentapeptide)-PP (M5PP) and MurNAc pentapeptide (M5) were established after digestion of LI with colicin M and hydrolysis of standard M5P with 20 mM HCl, respectively. The fluorogram on the right of Figure 7A shows that the major compound recovered from the 15 mM NH_4_OH eluate, after SPE of an incubation of LI with liver microsomal protein, co-migrates with standard M5P. These data demonstrate that solubilized rat liver microsomal protein cleaves bacterial LI to produce M5P, and that the presence of the terminal non-reducing GlcNAc residue is not essential for LII hydrolysis by solubilized rat liver microsomal protein. Next, the capacity of the microsomal preparation to hydrolyze a lipid II analog with a short polyprenol chain, LII(DAP)35 (see Table 1), was tested. The fluorograms shown in Figure 7B show that the short-chain lipid II analog is cleaved to yield GM5P, and that the rate of cleavage is similar to that noted for the longer chain, C55 analog, shown in Figure 3B.

### 2.4. The Biochemical Characteristics of GM5P Generation by Liver Microsomal Protein Extracts Are Similar to Those of the Mammalian DLODP Activity

The EDTA sensitivity of the lipid II cleavage reaction (Figure 4) and the fact that the reaction products of both LI and LII cleavage contain a single phosphate residue are consistent with the hypothesis that these structures are hydrolyzed by the previously characterized liver DLODP. In order to further substantiate this hypothesis, the biochemical characteristics of LII hydrolysis were studied in more detail. To do this, LII(DAP)35 was used as a substrate and, as shown in Figure 8A, when incubated for 4 h, GM5P production was linear over the range of protein quantities tested. When varying quantities of the substrate were incubated with 27.2 μg of protein for 4 h, it was found that GM5P production was saturable, and was ~75% maximal when 500 pmol LII were used (Figure 8B). Finally, when 500 pmol LII was incubated with 27.2 μg of protein, GM5P production was reasonably linear up until 4 h of incubation (Figure 8C). These reaction conditions were used to further characterize the characteristics of the GM5P-generating reaction. The detection of mammalian DLODP activity using a crude membrane preparation from HepG2 cells requires the presence of Co^2+^ [13]. While optimizing the detergent, salt, and Co^2+^ concentrations required to solubilize the rat liver microsomal DLODP activity, it was found that DLODP activity recovery was poor unless Co^2+^ was added to the solubilization buffer. Accordingly, microsomal proteins were solubilized in the presence of either Co^2+^, Mn^2+^, Ca^2+^, or Mg^2+^ before assaying the lipid II hydrolyzing activities in the corresponding cation-containing buffers. Data shown in Figure 9A demonstrate that GM5P production from LII(DAP)35 is strikingly increased when Co^2+^ is present in the solubilization and assay buffers. DLODP activity is blocked by solanesyl (Sol)-based compounds with the following order of efficacy: Sol-PP-GlcNAc_2_ > Sol-PP-GlcNAc > Sol-PP > Sol-P > solanesol [13]. As demonstrated by the data shown in Figure 9B, these compounds inhibit GM5P production from LII(DAP)35 with the same order of potency. Finally, DLODP activity is inhibited by sodium orthovanadate and Hg^2+^, but not by sodium fluoride [13], and, as shown in Figure 9C, GM5P production from LII(DAP)35 shows the same sensitivity to these reagents. Taken together, these data indicate that the same activities, or activities with closely related biochemical characteristics, generate GM5P from LII(DAP)35 and [^3^H]OSP from [^3^H]DLO.

## 3. Discussion

The SPE extraction of reaction mixtures using Vm-beads facilitated the rapid analysis of lipid II and its hydrolysis products in reaction mixtures containing liver microsomal proteins. Although this procedure was successful under the conditions employed here, its use in a routine assay requires caution because it relies on binding of the d-Ala–d-Ala motif to vancomycin. Even under our experimental conditions that include a protease inhibitor cocktail, it is possible that this motif can be destroyed, and that some substrate and product are, therefore, not able to be extracted by the Vm-beads. In fact, especially with longer incubation periods, significant increases in fluorescence were detected in the SPE flow-through fractions; however, the emission excitation spectra of this material were different to those expected of dansylated compounds. In addition, incubations were conducted in the presence of fragments of the bacterial lipid pentapeptide moiety, which might be expected to compete for peptidases and protect lipid II substrates and/or products from degradation. Although the acetylated tripeptide Ac-Lys–d-Ala–d-Ala binds efficiently to vancomycin, the dipeptide d-Ala–d-Ala (see Table 1) does not, and inclusion of the latter compound, at a concentration of 2 mM (200-fold above the amount of lipid II), in incubation mixtures did not increase the recovery of GM5P after 4-h incubations with the liver microsomal extracts (S.M., unpublished observations). Similarly, under the same experimental conditions, muramyl dipeptide (MDP, see Table 1), comprising the bacterial lipid I and II sugar/pentapeptide link region, had no effect on GM5P recovery (S.M., unpublished observations).

After SPE using VM-beads, the hydrolysis of lipid II was followed by quantitating the fluorescent hydrolysis products after TLC. Other readouts for lipid II hydrolysis that do not require the TLC step are possible. Firstly, as shown in Figure 4A, either the decrease in fluorescence associated with the 15 mM NH_4_OH/MeOH eluates, or the increase in fluorescence associated with the 15 mM NH_4_OH eluates could be used as a measure of lipid II cleavage.

SPE extraction of enzyme incubation mixtures using Vm-beads permitted direct characterization of lipid II hydrolysis products by TLC and allowed us to demonstrate that a preparation of solubilized mouse liver microsomal protein promotes the generation of monophosphoryl compounds from LI and LII. We did not determine whether any of the bacterial lipid analogs are better substrates than others. This type of kinetic study will have to await a pure enzyme preparation. Again, although it was shown that GM5P production from LII was protein- and time-dependent, as well as being saturable with respect to substrate, the kinetics of the reaction were not examined in more detail because of the crude nature of the enzyme preparation. Importantly, using solubilized liver microsomal proteins, we were able to show a close correlation between the biochemical characteristics of GM5P production and those previously reported for DLODP-mediated [^3^H]OSP generation from [^3^H]DLO using human hepatocellular carcinoma HepG2 cell membranes. Coupled with previously obtained data showing that bacterial lipid II inhibits [^3^H]OSP generation from [^3^H]DLO, our new data demonstrate that [^3^H]OSP and GM5P are generated, respectively, from [^3^H]DLO and LII by the same or a closely related liver microsomal activity.

To conclude, we report a simple SPE procedure that allows direct quantitation and characterization of the LII hydrolysis products that are generated by mammalian liver protein extracts, by either TLC or mass spectrometry. We demonstrate, for the first time, the hydrolysis of LII and LI to yield, respectively, GM5P and M5P by mammalian microsomal proteins. As the biochemical characteristics of this reaction are similar to those displayed by the reaction required for [^3^H]OSP generation from [^3^H]DLO, dansyl bacterial lipid I and II analogs are useful reagents with which to investigate mammalian and yeast DLODP and DLODP-like enzymes. This simple non-radioactive assay should allow biomedical workers to look for changes in DLODP-like activities in cells from patients with suspected glycosylation diseases. Defective DLODP activities may underlie as yet unrecognized disorders or modify disease severity in CDGs where truncated DLO are seen.

## 4. Materials and Methods

### 4.1. Reagents

d-Ala–d-Ala, Ac-Lys–d-Ala–d-Ala, sodium orthovanadate, vancomycin, HCl, NHS-Sepharose, and mammalian and yeast protease inhibitor cocktails were purchased from Sigma–Aldrich SARL (St Quentin Fallavier, France). Muramyl dipeptide (MDP, see Table 1) was from InvivoGen (Toulouse, France). Nonidet P-40 (NP-40) was obtained from Thermo-Scientific (Villebon-sur-Yvette, FR). Ultima Gold was from PerkinElmer Life Sciences (Zaventem, Belgium). TLC plates were from MERCK (Darmstadt, Germany).

### 4.2. Synthesis of Bacterial Lipids and Their Precursors and Dolichol-Linked Oligosaccharide Analogs

Fluorescently labeled dansyl(Lys)lipid II (Table 1; LII(Lys)55) was synthesized by chemi-enzymatic preparation of dansylated UDP-MurNAc pentapeptide [22], followed by its linkage to undecaprenyl phosphate and UDP-GlcNAc by membrane-bound MraY and MurG, respectively, to form lipid II. Lipid II was subsequently purified as previously described [23]. Uridinyl diphospho dansyl(DAP)pentapeptide (UDPM5, see Table 1), dansyl(DAP)lipid II (LII(DAP)55, see Table 1), and dansyl(DAP)35 lipid II (see Table 1; LII(DAP)35) were produced by enzymatic synthesis as described previously [20,24,25]. Dansyl phospho-MurNAc-pentapeptide (M5P, see Table 1) was prepared by treatment of dansyl UDPM5 with nucleotide pyrophosphatase [26,27]. Solanesyl-based DLO analogs were synthesized as previously reported [15].

### 4.3. Liver Microsomes and Protein Solubilization

Microsomes, prepared from rat liver as previously described [13] and stored at −150 °C, were resuspended at a concentration of 2 mg protein/mL in buffer containing 1 mM CoCl_2_, 500 mM NaCl. and 50 mM 2-(N-Morpholino)ethanesulfonic acid (MES), pH 6.0. After incubating for 1 h at 4 °C, the membranes were centrifuged at 100,000 gAv for 30 min. The resulting pellet was then suspended at a concentration of 1–2 mg protein/mL in buffer containing 0.25% NP-40, 1 mM CoCl_2_, 200 mM NaCl, and 50 mM MES, pH 6.0 (solubilization buffer, SB). After incubating for 1 h at 4 °C, the membranes were centrifuged at 100,000 gAv for 30 min. Determination of DLODP activity in the resulting supernatant and pellet revealed that 80–90% of the activity was solubilized under these conditions. The supernatants were stored at −150 °C.

### 4.4. DLODP Assay Using [^3^H]Man_5_GlcNAc_2_-PP-dolichol-2

Firstly, 30 μg of solubilized rat liver microsomal proteins was incubated with 15.0 × 10^3^ cpm [^3^H]Man_5_GlcNAc_2_-PP-dolichol [13] in SB in a total volume of 50 μL at 37 °C for 20 min. Reactions were stopped by the addition of 450 μL of ice-cold H_2_O and then loaded onto coupled 0.5-mL columns of Dowex 50-X2 (H^+^ form) and Dowex 1-X2 (acetate form). Subsequent to extensive washing of the two columns with H_2_O, [^3^H]oligosaccharyl phosphates were eluted from the Dowex 1-X2 columns with 5 mL of 3 M formic acid. After drying under vacuum, the radioactive material was taken up in 200 μL of H_2_O and quantitated by scintillation counting after mixing with 3 mL of scintillant.

### 4.5. Preparation of Vancomycin Beads

Vancomycin was covalently linked to NHS-Sepharose using the coupling protocol supplied with the NHS-Sepharose beads. Vancomycin HCl was dissolved (12.5 mg/mL) in 0.5 M NaHCO_3_ (pH 8.0) and one volume equivalent of this solution was added to two volume equivalents of beads and incubated with constant mixing for 3 h at room temperature. The beads were washed, alternately, with low- and high-pH buffers as described in the manufacturer’s protocol, and then stored as a 1:1 suspension in 50 mM Tris/HCl, pH 7.4, containing 0.05% sodium azide. To determine the coupling efficiency, unreacted vancomycin was separated from *N*-hydrosuccinimide (NHS), which is liberated during the coupling reaction, using C18 Sep Pak cartridges, and assayed spectrophotometrically (280 nm). Using this procedure, it was determined that 0.7 μmol vancomycin was coupled per mL of settled beads (Vm-beads).

### 4.6. Assay of Bacterial Lipid II Hydrolysis

Dansylated bacterial lipid II analogs were aliquoted into 1.5-mL reaction tubes and, after removal of organic solvents under vacuum, were suspended in 5 μL of SB. After addition of protein solubilized in SB, and other additions described in the figure legends, the mixtures were incubated at 37 °C for 0–16 h. Reactions were stopped by diluting 10-fold in 50 mM Tris/HCl, pH 7.4, containing 200 mM NaCl. After adding 80 μL of the 1:1 suspension of Vm-beads, the mixtures were incubated with constant rotation for 1 h at 4 °C. The contents of the reaction tubes were then transferred to 0.8 × 4-cm polypropylene columns (Poly-Prep Chromatography Columns, BIO-RAD) and the Vm-beads were washed sequentially with 6 × 200 μL of Vm-bead wash buffer (50 mM Tris/HCl, pH 7.4, 200 mM NaCl, 0.02% NP-40), 6 × 200 μL of H_2_O, 6 × 200 μL of 15 mM NH_4_OH, and finally 6 × 200 μL of 15 mM NH_4_OH in MeOH. Fractions were collected into 96-well flat black plates (Costar) and fluorescence (excitation 350 nm, emission 530 nm) was measured using an Infinite M200 microplate reader (Tecan). Where indicated, the 15 mM NH_4_OH and 15 mM NH_4_OH/MeOH eluates were dried under vacuum and analyzed by thin-layer chromatography using silica-coated HPTLC plates (Merck) developed in *n*-propanol/ammonia/water (6:3:1) [20]. After drying the plates, fluorescent compounds were detected using a Quantum Gel Imager (Vilber-Lourmat), and quantitated by densitometric scanning using the Bio 1D software. Where indicated, the amount of product generated was estimated by calibrating the TLC system with 0–200 pmol standard UDPM5 (Table 1). This compound was also used in order to calculate the recovery of water-soluble d-Ala–d-Ala-containing compounds after SPE of standard incubation mixtures.

### 4.7. Digestion of Bacterial Lipids with Colicin M

Lipids I or II (500 pmol) were incubated with 3 µg of recombinant colicin M [20] in 50 µL of a buffer containing 50 mM TrisHCl pH 7.4, 150 mM NaCl, 20 mM MgCl_2_, and 0.12% NP-40 for 16 h at 37 °C [20]. The reaction mixtures then underwent SPE using Vm-beads as described above.

### 4.8. Mass Spectrometry

Samples were infused in an Orbitrap Fusion Tribrid (Thermo Scientific) coupled to a TriVersa NanoMate nano-electrospray ion source (Advion). Compounds were analyzed in the Orbitrap cell, in negative ion mode, at 1.4 kV, in full ion scan mode, at a resolution of 30,000 (at *m*/*z* 200), with a mass range of *m*/*z* 500–1500, an automatic gain control target of 2 × 10^5^, and a maximum ion accumulation time of 100 ms. MS/MS data were acquired in the ion trap and obtained by high collision-induced dissociation (HCD) activation with a collisional energy of 30%, and a quadrupole isolation window of 1.6 Da. The maximum ion accumulation times were set to 100 ms for MS acquisition and MS/MS acquisitions. The isotopolog cluster of the GM5P compound obtained in MS was compared with the theoretical cluster that was obtained using the Metabolite Identification via Database Searching algorithm [21].

## Figures and Tables

**Figure 1 molecules-24-02135-f001:**
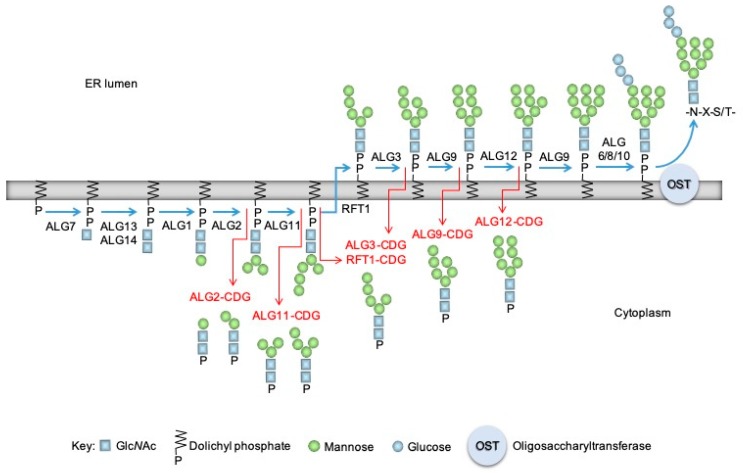
The dolichol cycle, *N*-glycan biosynthesis, and the production of oligosaccharylphosphates in congenital disorders of glycosylation. Dolichyl phosphate (left) is glycosylated on the cytoplasmic face of the ER to yield Man_5_GlcNAc_2_-PP-dolichol which is flipped into the ER and further elongated to yield mature Glc_3_Man_9_GlcNAc_2_-PP-dolichol (DLO; blue arrows). Gene names of the proteins required for some of the major steps are indicated next to the blue arrows. Oligosaccharyltransferase (OST) transfers the oligosaccharide of mature DLO onto nascent polypeptides (N-X-S/T). In some congenital disorders of glycosylation where genes of the dolichol cycle are mutated, increased amounts of cytoplasmic oligosaccharyl phosphates (OSP), indicated by red arrows, are detected. By way of example, in ALG12-CDG [7] the Alg12p mannosyltransferase that adds the eighth mannose residue to the growing DLO is defective, and both a truncated DLO and an OSP containing seven residues of mannose accumulate [8]. Furthermore, and not shown here, in both PMM2-CDG, in which phosphomannomutase encoded by the *PMM2* gene is deficient, and glucose starvation, nucleotide sugars required by glycosyltransferases of the dolichol cycle are limiting, and truncated DLO and the corresponding OSP are observed [8,9,10].

**Figure 2 molecules-24-02135-f002:**
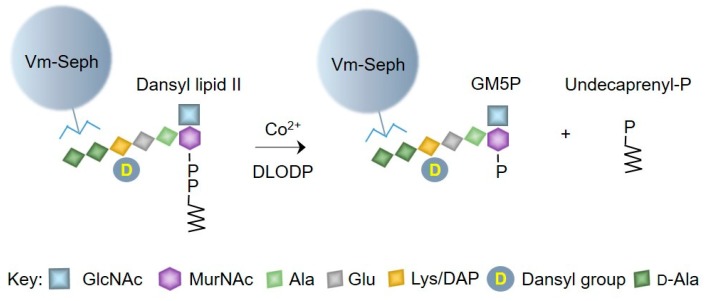
DLO diphosphatase (DLODP) assay strategy involving capture of lipid II and its potential hydrolysis products on vancomycin–Sepharose affinity beads. It is hypothesized that the previously described Co^2+^-dependent DLODP activity can hydrolyze dansylated bacterial lipid II to yield undecaprenyl phosphate and GlcNAcβ1,4MurNAc(dansylpentapeptide)-P (GM5P). Both lipid II and the expected hydrolysis product contain a d-Ala–d-Ala motif that binds to vancomycin and could potentially be purified from reaction mixtures using solid-phase extraction with vancomycin covalently linked to Sepharose beads (Vm-Seph).

**Figure 3 molecules-24-02135-f003:**
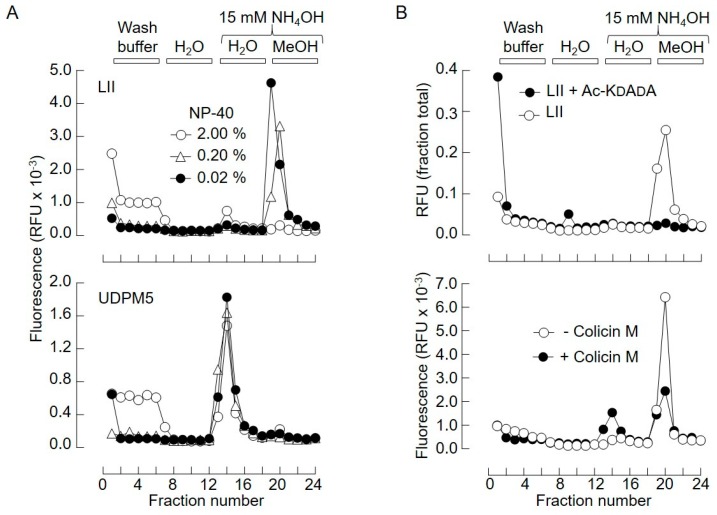
(**A**) Solid-phase extraction of lipid II and uridinyl diphospho (UDP)-MurNAc pentapeptide (UDPM5) from detergent-containing mixtures. (**B**) 500 pmol LII(Lys)55 (upper panel) or UDPM5 (lower panel) was incubated with Vm-beads in Vm wash buffer containing the indicated amounts of nonyl phenoxypolyethoxylethanol (NP-40). The beads were then washed with the indicated solutions and fractions were collected into 96-well plates and monitored for fluorescence. (B, upper panel) 500 pmol LII(Lys)55 was incubated with Vm-beads in the presence of 1.5 µmol (3000-fold excess over lipid II and ~50-fold excess over vancomycin attached to beads) acetyl-Lys-D-Ala-D-Ala (Ac-KDADA). (B, lower panel) 500 pmol LII(Lys)55 was incubated in either the absence or presence of colicin M as described in Section 4. The reaction mixtures were subjected to solid-phase extraction as described above.

**Figure 4 molecules-24-02135-f004:**
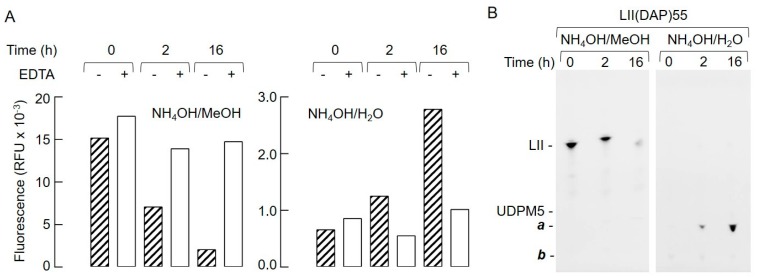
Hydrolysis of lipid II by rat liver microsomal proteins. (**A**) LII(DAP)55 (500 pmol) was incubated with detergent-solubilized rat liver microsomal proteins, as described in Section 4, for the indicated times in either the absence or presence of 10 mM ethylenediaminetetraacetic acid (EDTA). After solid-phase extraction (SPE) using Vm-beads as described for Figure 1, the fluorescence values associated with the fractions eluted with 15 mM NH_4_OH in methanol (left panel) and 15 mM NH_4_OH in H_2_O (right panel) were summed. (**B**) LII(DAP)55 (500 pmol) was incubated as described above for the indicated times in the absence of EDTA. After SPE using Vm-beads, the 15 mM NH_4_OH/methanol and 15 mM NH_4_OH/H_2_O eluates were dried down and analyzed by thin-layer chromatography (TLC) as described in Section 4. The migration positions of LII(DAP)55 (LII) and uridine diphospho MurNAc pentapetide (UDPM5) are shown. The components **a** and **b** are discussed in the text.

**Figure 5 molecules-24-02135-f005:**
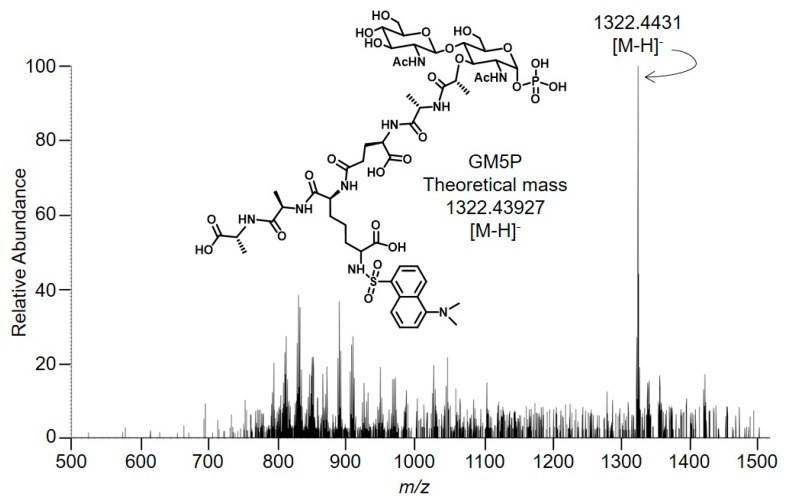
Characterization of Lipid II digestion products by mass spectrometry. Five reaction mixtures, each containing 500 pmol LII(DAP)35 and 27.2 μg rat liver microsomal protein, were incubated for 16 h as described in Section 4. After SPE with Vm-beads, the 15 mM NH_4_OH/H_2_O eluates were dried under vacuum and taken up in 25 μL of H_2_O. This material was directly infused, via a nano-electrospray ion source, into an Orbitrap Fusion mass spectrometer. The spectrum obtained in negative ion mode and the expected digestion product (GM5P) are represented.

**Figure 6 molecules-24-02135-f006:**
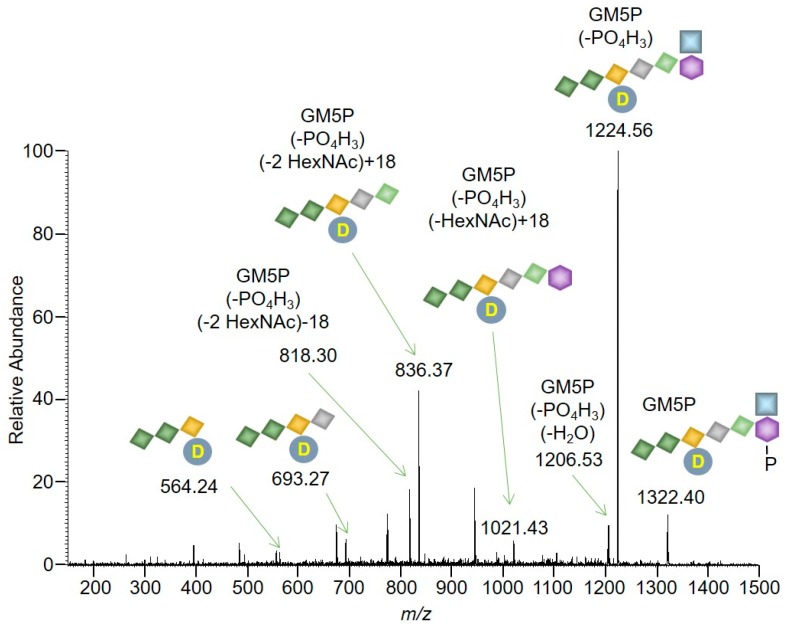
Characterization of Lipid II digestion products by mass spectrometry. The MS/MS spectrum acquired in negative mode and in an ion trap was generated as described in Section 4. Signals compatible with fragments generated from GM5P are indicated. Losses (*m*/*z* = −18) or adducts (*m*/*z* = +18) of H_2_O molecules are indicated in the spectrum. The symbols used are defined in Figure 2.

**Figure 7 molecules-24-02135-f007:**
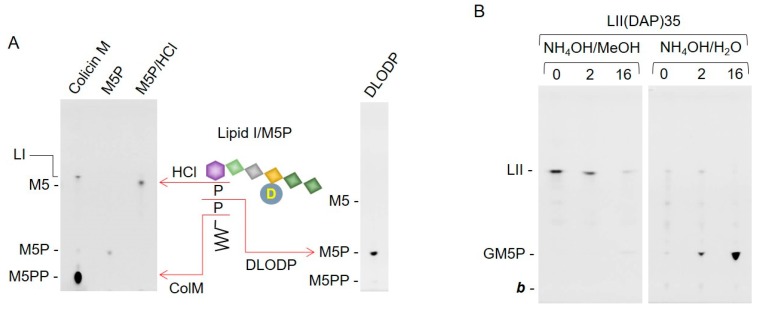
Lipid I and a short-chain LII analog are also hydrolyzed by rat liver microsomal protein. (**A**) LI (500 pmol) was digested with colicin M as described in Section 4, and 100 pmol MurNAc(pentapeptide)-P was treated (M5P/HCl), or not (M5P), with 20 mM HCl for 45 min at 100 °C. After SPE extraction, the 15 mM NH_4_OH eluates were dried and analyzed by TLC (left fluorogram). LI (500 pmol) was digested with detergent-solubilized rat liver microsomal proteins (27.2 μg), as described in Section 4, for 16 h. After SPE, the 15 mM NH_4_OH eluates were dried under vacuum and analyzed by TLC (DLODP, right fluorogram). The migration positions of lipid I, MurNAc pentapeptide (M5), MurNAc(pentapeptide)-P (M5P), and MurNAc(pentapeptide)-PP (M5PP) are indicated to the left of the fluorograms. (**B**) The hydrolysis of LII(DAP)35 (500 pmol) was examined as described for LII(DAP)55 in the legend to Figure 4.

**Figure 8 molecules-24-02135-f008:**
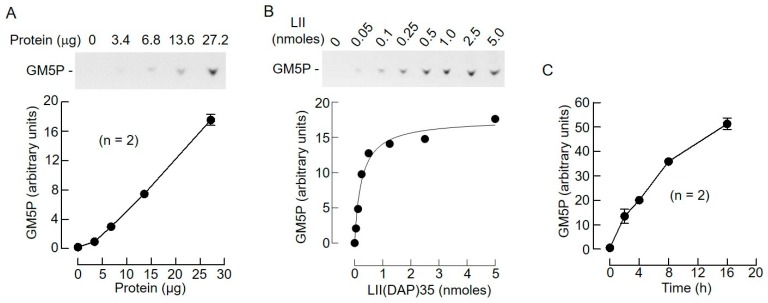
Biochemical characteristics of rat liver lipid II hydrolyzing activity. (**A**) LII(DAP)35 (500 pmol) was incubated with different quantities of detergent-solubilized rat liver microsomal proteins for 4 h. After SPE, the 15 mM NH_4_OH/H_2_O eluates were dried under vacuum and analyzed by TLC. The resulting fluorograms were scanned and GM5P was quantitated as described in Section 4. The means and SDs of duplicate incubations were generated using Prism software. One of the fluorograms is shown. (**B**) 27.2 μg of detergent-solubilized rat liver microsomal proteins was incubated for 4 h with increasing quantities of LII(DAP)35, (LII nmol), and GM5P was quantitated as described above. A single experiment was performed. (**C**) LII(DAP)35 (500 pmol) was incubated with 27.2 μg of detergent-solubilized rat liver microsomal proteins for the indicated times. GM5P was quantitated as described above.

**Figure 9 molecules-24-02135-f009:**
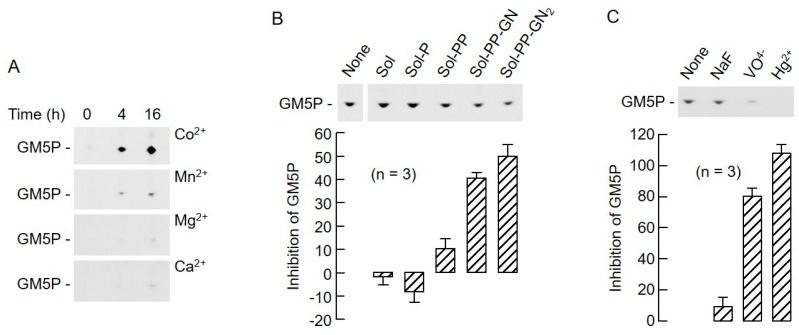
Biochemical profile of LII hydrolyzing activity. (**A**) Rat liver microsomes were solubilized in buffers containing 1 mM CoCl_2_ (Co^2+^), MnCl_2_ (Mn^2+^), MgCl_2_ (Mg^2+^), or CaCl_2_ (Ca^2+^). The extracts (20 μg) were incubated with 500 pmol LII(DAP)35 for the indicated times, and, after SPE, the 15 mM NH_4_OH eluates were dried down and analyzed by TLC. The migration position of GM5P is indicated to the left of the fluorograms. (**B**) LII(DAP)35 (500 pmol) was incubated with 27.2 μg of detergent-solubilized rat liver microsomal proteins in the presence of 2.5 nmol solanesol (Sol), solanesyl-P (Sol-P), solanesyl-PP (Sol-PP), solanesyl-PP-GlcNAc (Sol-PP-GN), or solanesyl-PP-GlcNAc_2_ (Sol-PP-GN_2_) for 4 h as described in Section 4. GM5P was quantitated as detailed above, and is expressed as a percentage of the quantity of GM5P generated in the absence of the solanesyl compounds (None). (**C**) Incubations were conducted exactly as described for (B) in the absence of additions (None) or presence of either 50 mM sodium fluoride (NaF), 5 mM sodium orthovanadate (VO^4−^), or 1 mM mercury chloride (Hg^2+^). GM5P was quantitated and expressed as for (B).

**Table 1 molecules-24-02135-t001:** Bacterial lipid I and II analogs and fragments used in this study.

Abbreviation	Compound	Abbreviation	Compound
LII(Lys)55	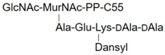	UDPM5	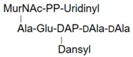
LII(DAP)55	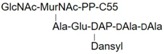	M5P	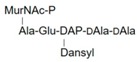
LII(DAP)35	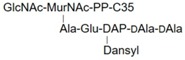	GM5P	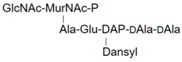
LI	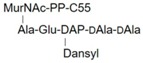	Ac-KdAdA	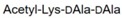
MDP	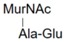	dAdA	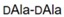

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
