# Peer review of "Bacterial Lipid II Analogs: Novel In Vitro Substrates for Mammalian Oligosaccharyl Diphosphodolichol Diphosphatase (DLODP) Activities"

_molecules, 2019, doi:10.3390/molecules24112135_

Round 1
Reviewer 1 Report
Present manuscript reports a new method to measure activity of DLODP which might be an essential enzyme in mammalian sugar chain regulation using lipid II as its substrate. The method utilizes vancomycin affinity beads to selectively recover the lipid II substrate. Although this method inherent limitation, it still is a good idea and useful to analyze the activity of DLODP. The enzymatic characterization of DLODP here is preliminary, further study using purified enzyme is warranted using this method. The manuscript is written well, however, if the authors add a schematic figure in introduction to show sugar chain biosynthesis in relation to the experimental design to this work, it would be much easier to read for general readers who are not familiar to the subject. Overall this manuscript should be accepted for publication in Molecules.
Author Response
The manuscript is written well, however, if the authors add a schematic figure in introduction to show sugar chain biosynthesis in relation to the experimental design to this work, it would be much easier to read for general readers who are not familiar to the subject.
Reply
We have now added a schematic figure in the introduction. The introduction has been slightly modified to accommodate this addition.
Reviewer 2 Report
This article concerns the biochemical detection of an enzyme activity involved in the dolichol-phosphate pathway, which is not readily available at present and is relevant in the study of N-glycan biosynthesis, in particular of some congenital disorders of glycosylation (CDG).
The topic is interesting, the results convincing and well presented. However, the article claims to provide a new tool in the field potentially suitable to readers belonging to a biomedical area of expertise. By this point of view, I suggest to improve the introduction and/or discussion with clear examples showing specific experimental fields of application, including specific CDGs and related biochemical questions.
The authors are also required to turn CDG nomenclature to the current recommendations (CDG-I and CDG-I subtypes are not acceptable), including more recent related references.
Author Response
The topic is interesting, the results convincing and well presented. However, the article claims to provide a new tool in the field potentially suitable to readers belonging to a biomedical area of expertise. By this point of view, I suggest to improve the introduction and/or discussion with clear examples showing specific experimental fields of application, including specific CDGs and related biochemical questions.
Reply
We have now added a sentence at the end of the discussion (lines 337 - 340) indicating how this assay could be used to further understanding of glycosylation diseases.
The authors are also required to turn CDG nomenclature to the current recommendations (CDG-I and CDG-I subtypes are not acceptable), including more recent related references.
Reply
The term CDG-I has now been removed and we now use exclusively the new nomenclature. The changes occur in the introduction and the new Figure 1 and it's legend. In addition we have replaced the two CDG references (3 and 4) with two more up to date references.